# High performance n-type $Ag_2Se$ film on nylon membrane for flexible thermoelectric power generator

Yufei Ding[1], Yang Qiu[2], Kefeng Cai[1], Qin Yao[3], Song Chen[4], Lidong Chen[3] & Jiaqing He[2]

Researches on flexible thermoelectric materials usually focus on conducting polymers and conducting polymer-based composites; however, it is a great challenge to obtain high thermoelectric properties comparable to inorganic counterparts. Here, we report an n-type $Ag_2Se$ film on flexible nylon membrane with an ultrahigh power factor ~987.4 ± 104.1 $\mu Wm^{-1}K^{-2}$ at 300 K and an excellent flexibility (93% of the original electrical conductivity retention after 1000 bending cycles around a 8-mm diameter rod). The flexibility is attributed to a synergetic effect of the nylon membrane and the $Ag_2Se$ film intertwined with numerous high-aspect-ratio $Ag_2Se$ grains. A thermoelectric prototype composed of 4-leg of the $Ag_2Se$ film generates a voltage and a maximum power of 18 mV and 460 nW, respectively, at a temperature difference of 30 K. This work opens opportunities of searching for high performance thermoelectric film for flexible thermoelectric devices.

[1] Key Laboratory of Advanced Civil Engineering Materials, Ministry of Education, School of Materials Science and Engineering, Tongji University, 4800 Caoan Road, 201804 Shanghai, China. [2] Physics Department, Southern University of Science and Technology, 1088 XueYuan Avenue, 518055 Shenzhen, China. [3] State Key Laboratory of High Performance Ceramics and Superfine Microstructure, Shanghai Institute of Ceramics, Chinese Academy of Science, 200050 Shanghai, China. [4] School of Materials Science and Engineering, Fujian University of Technology, 350108 Fuzhou, China. Correspondence and requests for materials should be addressed to K.C. (email: kfcai@tongji.edu.cn) or to L.C. (email: cld@mail.sic.ac.cn) or to J.H. (email: he.jq@sustc.edu.cn)

N owadays, the explosive growth of wearable devices has stimulated the development of materials which can power the devices with the energy harvesting from the human body[1]. Flexible thermoelectric (TE) materials as the promising energy-harvesting materials, which can directly convert heat into electricity or vice versa and realize the self-power for wearable devices from the temperature difference between the skin and the ambient environment, have attracted increasing attention[1–8]. The TE performance of a materials is evaluated by the dimensionless figure of merit, $ZT = \alpha^2\sigma T\kappa^{-1}$, where $\alpha$, $\sigma$, and $\kappa$ are the Seebeck coefficient, electrical conductivity, and thermal conductivity of the material, respectively, and $T$ is the absolute temperature.

Until now, the research on flexible TE materials mainly focuses on conducting polymers (CPs) and CP-based composite materials[9–12]; although much progress has been made[4,13–16], their ZT values are still incompatible to those of the inorganic TE materials. Moreover, the CP-related materials are mainly p-type while their n-type counterpart is still lacking. For the completion of a high efficiency flexible TE module, high performance of n-type materials are very desirable.

Besides the CPs, insulating polymers are also employed for forming TE composites with inorganic TE materials. For example, most recently, Hou et al.[17] prepared p-type $Bi_{0.5}Sb_{1.5}Te_3$ and epoxy resin composite thick film by hot-pressing (623 K, 4 MPa) and the film shows a high power factor (PF = $\alpha^2\sigma$) of 840 $\mu$W m$^{-1}$ K$^{-2}$. Recently, flexible substrate, such as polyimide, fiber or paper, has been used to support inorganic materials for preparing high performance and flexible TE materials[17–22]. For instance, Gao et al.[21] reported a glass-fiber-aided cold-pressing method for flexible n-type $Ag_2Te$ films on copy paper and the power factor value was up to 85 $\mu$W m$^{-1}$ K$^{-2}$ at 300 K. Choi et al.[22] prepared n-type HgSe nanocrystal thin film by spin-coating HgSe nanocrystal solution on a plastic substrate and the film showing a maximum power factor of 550 $\mu$W m$^{-1}$ K$^{-2}$ at 300 K. Jin et al.[23] deposited $Bi_2Te_3$ thick film on a cellulose fibers paper via an unbalanced magnetron sputtering technique and the composite film exhibits good flexibility and a power factor of ~250 $\mu$W m$^{-1}$ K$^{-2}$ at room temperature.

$\beta$-$Ag_2Se$ is a narrow band semiconductor with an energy gap $E_g = 0.07$ eV at 0 K and it transforms into a cubic superionic conductor ($\alpha$-$Ag_2Se$) around 407 K. $\beta$-$Ag_2Se$, which exhibits high electrical conductivity and low thermal conductivity, has the great potential for n-type TE material near room temperature. Several groups have reported the TE performance of $Ag_2Se$[24–26]. For instance, Ferhat et al.[27] prepared $\beta$-$Ag_2Se$ bulks by a direct-reaction of the element Ag and Se in evacuated quartz tubes at 1273 K and the maximum power factor of the material was about 3500 $\mu$W m$^{-1}$ K$^{-2}$, which is similar to that of the state-of-art material at room temperature. Most recently, Perez-Taborda et al.[28] deposited $Ag_2Se$ films on glass substrates via pulsed hybrid reactive magnetron sputtering and the films showing a high power factor ~2440 $\mu$Wm$^{-1}$ K$^{-2}$ at room temperature. Nevertheless, the films are with relatively high cost since an expensive facility is used.

Although $Ag_2Se$ materials with excellent TE performance at room temperature have been reported, they are all non-flexible. In this work, we developed a facile strategy to prepare n-type flexible $Ag_2Se$ film on a nylon membrane. The $Ag_2Se$ film showed a very high power factor of 987.4 $\mu$Wm$^{-1}$K$^{-2}$ at 300 K, which is one of the best values reported for flexible n-type materials and even comparable to that of some high-ZT inorganic bulk materials at high temperatures, such as SnSe (~900 $\mu$Wm$^{-1}$K$^{-2}$ at 773 K)[29] and $Cu_{2-x}Se$ (1200 $\mu$Wm$^{-1}$K$^{-2}$ at 1000 K)[30].

## Results
### Characterization of $Ag_2Se$ film.
XRD analysis of the film reveals that all the XRD peaks (Fig. 1a) can be indexed to $\beta$-$Ag_2Se$ phase

(JCPDS No. 24-1041). The XRD peaks for the $Ag_2Se$ film are stronger than those for the $Ag_2Se$ nanowires (NWs) (Supplementary Fig. 3a), and especially the (002) and (004) plane peaks become particularly strong, indicating increase of crystallinity and a large number of the $Ag_2Se$ grains preferentially grown along the (00l) plane[31]. The thickness of the $Ag_2Se$ film is about 10 $\mu$m (Supplementary Fig. 5). After hot pressing, the $Ag_2Se$ NWs with diameter of ~65 nm and length of a few micrometers are sintered into a network-like film with numerous submicron pores (Fig. 1b, c).

To investigate more detailed microstructure of the film, transmission electron microscope (TEM) sample was prepared by FIB and studied by high-angle annular dark field scanning TEM (HAADF-STEM). Figure 1d shows an overview HAADF-STEM image. Although pores ranging from dozens of nanometers to several-hundred nanometers and microgaps indicated by arrows can be observed, generally interior of the film is also dense. Figure 1e shows a typical STEM image. The white dots in Fig. 1e are Pt introduced during the TEM sample preparation. Figure 1f is a FFT pattern corresponding to Fig. 1e. Figure 1g is a typical high-resolution STEM (HRSTEM) image, in which the dot-lines shows grain boundaries, showing well sintered of the $Ag_2Se$ nanograins with different orientations and defects (edge dislocations and stacking faults). XPS analyses (see Supplementary Fig. 6) indicate that the film is stable and no oxidation occurred during the processing.

### TE properties of the $Ag_2Se$ film.
Figure 2a, b exhibit the TE properties and Hall measurement results of the $Ag_2Se$ film from 300 to 453 K, respectively. The Seebeck coefficient of the film at 300 K is about $-140.7$ $\mu$V K$^{-1}$, indicating n-type conduction. As temperature increases, the absolute Seebeck coefficient shows a decrease tendency and it decreases rapidly when the temperature increases from 393 to 423 K. The electrical conductivity of the film is ~497 S cm$^{-1}$ at 300 K, and it increases with the temperature increasing from 300 to 393 K and also decreases rapidly when the temperature decreases from 393 to 423 K. The temperature dependence of the Seebeck coefficient and electrical conductivity of the film can be understood from the Hall measurement results (see Fig. 2b), since the Seebeck coefficient is inversely proportional to the power of 2/3 of the carrier concentration[9] and the electrical conductivity, carrier concentration ($n$) and carrier mobility ($\mu$) have the relation: $\sigma = ne\mu$.

However, the change tendency above 393 K of the electrical conductivity of the film is not as that for the $Ag_2Se$ film reported in ref. 28, which is mainly because the carrier mobility of our film drops substantially (from ~800 to 400 cm$^2$ V$^{-1}$ s$^{-1}$) while that of the latter only drops from ~600 to 500 cm$^2$ V$^{-1}$ s$^{-1}$. The temperature dependence of carrier concentration and carrier mobility of the film is similar to that of the bulk $Ag_2Se$ reported in ref. 24, and it suddenly changes around 400 K which is due to the phase transformation from $\beta$-$Ag_2Se$ to $\alpha$-$Ag_2Se$ around 407 K. The carrier concentration first increases with increasing temperature, which is due to thermal excitation of carriers (the concurrent decrease in the Seebeck coefficient of the film suggests that the additional carriers are holes). When the temperature is beyond the phase transition temperature (407 K), the $Ag_2Se$ becomes a superionic conductor, and then the carrier concentration almost keeps constant. The mobility suddenly drops significantly across the superionic transition, the reason for which is that the Ag ions in superionic $Ag_2Se$ move freely and tend to scatter electrons more efficiently than a static lattice[24].

As a result, as the temperature increases, the power factor value also shows a similar change tendency to that of electrical conductivity: increases from 987.4 at 300 K to 1448.1 $\mu$Wm$^{-1}$ K$^{-2}$ at 393 K, then rapidly decreases to 569.2 $\mu$Wm$^{-1}$ K$^{-2}$ at ~423 K.

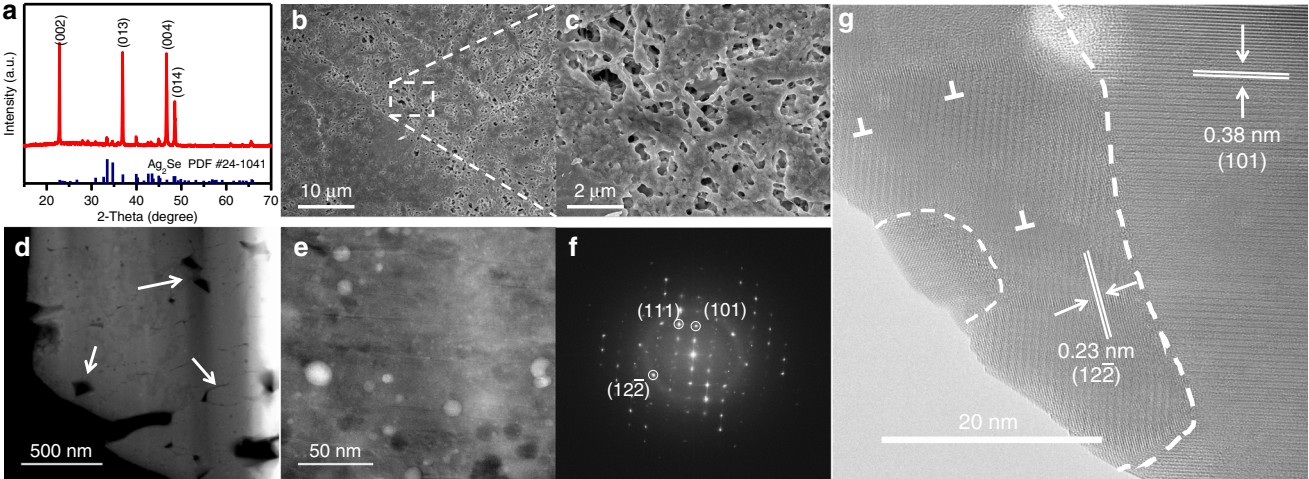

**Fig. 1** Characterization of the Ag$_2$Se film. **a** XRD pattern of the Ag$_2$Se film. **b** Low magnification surface FESEM image of the Ag$_2$Se film. **c** High magnification surface FESEM image of the Ag$_2$Se film. **d** Overview HAADF-STEM image. **e** Typical STEM image. **f** FFT image corresponding to **e**. **g** Typical HRSTEM image

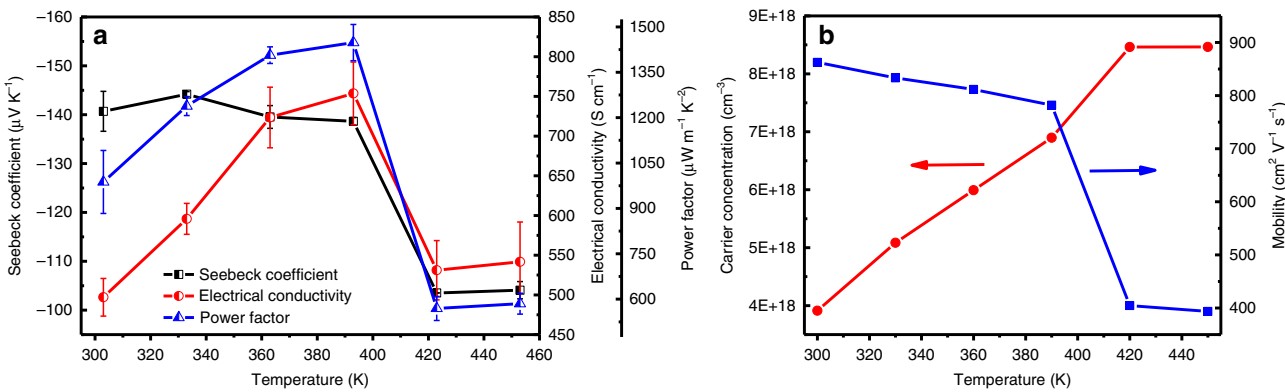

**Fig. 2** In-plane TE properties of the film. **a** Temperature dependence of Seebeck coefficient, electrical conductivity and power factor for the Ag$_2$Se film (Each point shows the standard deviations from two independent measurements). **b** Temperature dependence of carrier concentration and mobility for the Ag$_2$Se film

The power factor value at 300 K is among the best n-type flexible TE materials (See Table 1). However, it is still much lower than the best value of the bulk Ag$_2$Se with optimal carrier concentration reported in ref. [27]. In order to understand the mechanism, we hot pressed a bulk sample without nylon. The bulk sample, with similar relative density and its grains without preferential growth direction (see Supplementary Fig. 8), has a power factor of ~1340 μWm$^{-1}$ K$^{-2}$ at room temperature (see Supplementary Fig. 9). Hence, we deduce that the main reasons for the film with lower power factor are: the film with high porosity, its grains preferentially grown along (00$l$), and its carrier concentration not optimized.

As it is hardly to separate the Ag$_2$Se film from the nylon membrane without destroy the film, the in-plane thermal conductivity of the Ag$_2$Se film cannot be provided here (the in-plane thermal conductivity of the Ag$_2$Se film on nylon membrane was measured to be 0.449 Wm$^{-1}$K$^{-1}$, see Supplementary Table 1). However, it is deduced as follows: A $\kappa_e$ (=$L\sigma T$) of 0.268 Wm$^{-1}$ K$^{-1}$ is estimated for our film by using the same Lorenz number ($L$ = $1.8 \times 10^{-8}$ WΩK$^{-2}$) as used in ref. [28]. As our Ag$_2$Se film contains pores with sizes ranging from ~20 nm to submicrometers (see Supplementary Fig. 7 and Fig. 1c), nanograins (see Fig. 1g) and a

hetero-interface between the Ag$_2$Se film and nylon membrane (see Fig. 3b and Supplementary Fig. 5), disorder in terms of nanocrystallite boundaries, microstructural defects (dislocations and amorphous regions), heat-carrying phonons in a wide spectrum of wavelengths can be scattered. The microstructural defects, such as dislocations (a few nm) and heterointerfaces, are effective in scattering the short wavelength phonons, and the nanocrystallite boundaries and residual nanopores (~20–200 nm) play an effective role in scattering the mid-to-high wavelength phonons[32–34], thus the lattice contribution $\kappa_l$ will be lower than that ($\kappa_l \sim 0.21$ W m$^{-1}$ K$^{-1}$) of the Ag$_2$Se film consisting of micrograins in ref. [28], which suggests that the in-plane thermal conductivity ($\kappa_e + \kappa_l$) of our film will be lower than 0.478 W m$^{-1}$ K$^{-1}$. Hence, the ZT value at 300 K of our film is estimated to be ~0.6.

The TE properties of a β-Ag$_2$Se single crystal from 160 to 300 K are calculated using the first-principle density functional theory (see details in Supplementary Fig. 10 and Supplementary Note 5)[35]. The calculation shows that the power factor along the b-axis (($0l0$) direction is about 2465 μW m$^{-1}$ K$^{-2}$ at 300 K, which is about two orders of magnitude higher than that along other two axes. Recall that the present film is preferentially grown along (00$l$) direction (see Fig. 1a) and we hereinbefore deduced that the Ag$_2$Se grains

**Table 1 Comparison of TE performance of flexible n-type TE materials at room temperature**

| Materials | $\alpha$ ($\mu$V K$^{-1}$) | $\sigma$ (S cm$^{-1}$) | PF ($\mu$W m$^{-1}$ K$^{-2}$) | Ref. |
|---|---|---|---|---|
| Ag$_2$Te/copy-paper | −100 | 85 | 85 | 21 |
| Cu-doped Bi$_2$Se$_3$/PVDF | −84 | 146 | 103.2 | 43 |
| Ni NWs/PVDF | −20.6 | 4700 | 200 | 44 |
| Bi$_2$Te$_3$/Cellulose fiber | −130 | 148 | 250 | 23 |
| C$_{60}$/TiS$_2$ | −101 | 390 | 400 | 40 |
| HgSe | −518 | 20 | 550 | 22 |
| TiS$_2$[tetrabutylammonium]$_{0.013}$[hexylammonium]$_{0.019}$ | −150 | 400 | 904 | 45 |
| Ag$_2$Se/Nylon membrane | −140 | 497 | 987 | This work |

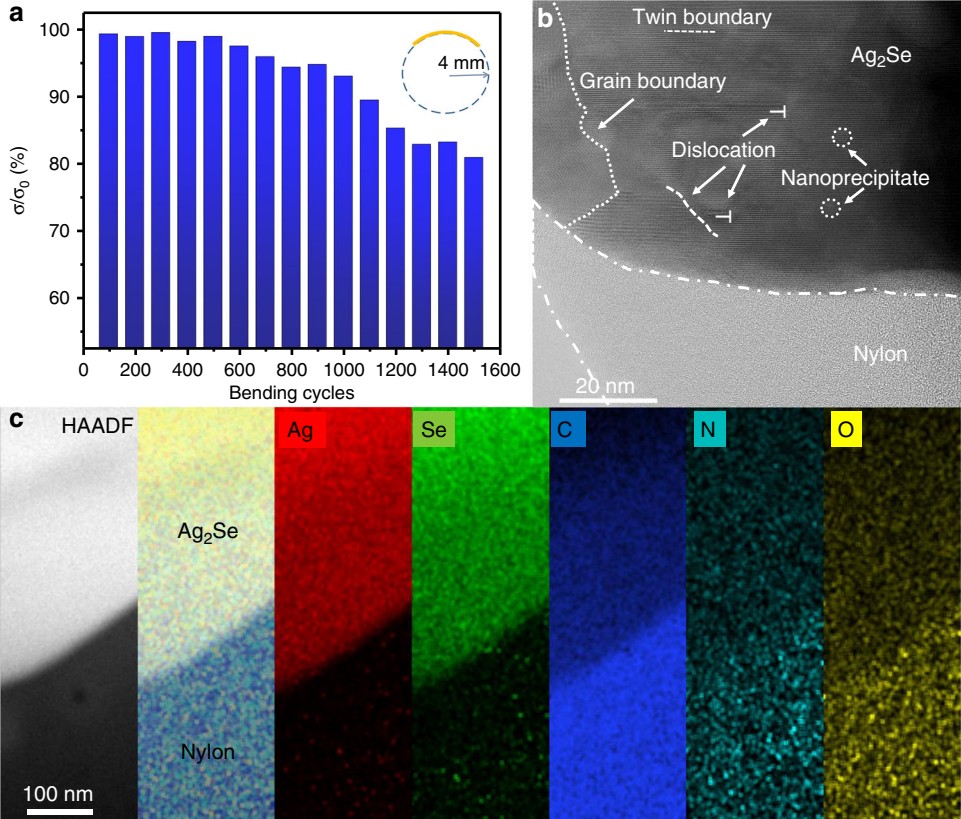

**Fig. 3** Flexibility of the film and evaluation of the heterointerface. **a** The ratio of electrical conductivity of the film before and after bending as a function of bending cycles. **b** HRSTEM image showing good combination between the Ag$_2$Se film and nylon membrane. **c** From left to right: HAADF image of a heterointerface between a Ag$_2$Se grain and nylon, overall corresponding EDS image and EDS image of elemental Ag, Se, C, N, and O

grown along (00*l*) direction is not good to TE properties, which agrees with the calculation results. This implies that the TE properties of our Ag$_2$Se film can be further improved via tuning the orientation of the Ag$_2$Se grains, increasing the density and optimizing the carrier concentration.

**Flexibility of the Ag$_2$Se film**. In order to test the flexibility of the Ag$_2$Se film, a bending test was applied around a rod with a diameter of 8 mm. Figure 3a demonstrates the ratio ($\sigma/\sigma_0$) of the electrical conductivity with or without bending with different cycles. The electrical conductivity decreases slowly with the increasing of bending cycles. About 93 and 80% of the initial electrical conductivity are maintained after 1000 and 1500 bending cycles, respectively. Compared with the n-type Ag$_2$Te films on copy paper[21], the n-type Bi$_2$Te$_3$ thick film on a cellulose

fibers paper[23], and p-type PEDOT/Bi$_2$Te$_3$ hybrid films with monodispersed and periodic Bi$_2$Te$_3$ nanophase fabricated via a very complicated process[16], our Ag$_2$Se film on nylon membrane shows a better flexibility (see Supplementary Table 2).

In order to better understand the excellent flexibility of the film. We deliberately examined the details near the interface between the Ag$_2$Se film and the nylon membrane. We found that the Ag$_2$Se grains are well combined with the amorphous nylon membrane (Fig. 3b), indicating that there is a good bonding between them, resulted from the hot pressing, which is good for flexibility[21]. In Fig. 3c, from left to right is a HAADF-STEM image near a heterointerface, overall corresponding EDS image, elemental EDS images of Ag, Se, C, N, and O, respectively. The elements of C, N, and O are attributed to the CONH group of the nylon. It is seen from the EDS images of elemental Ag and Se that

a small amount of these two elements are detected in the nylon membrane. This is because the nylon membrane is porous (pore size ~200 nm), and some tips of the $Ag_2Se$ nanowires may penetrate into the pores during the filtration and they bonded together during the hot pressing.

Therefore, the good flexibility of the $Ag_2Se$ film on nylon membrane can be explained by the following reasons. Firstly, as known to all, nylon has an intrinsic excellent flexibility. Secondly, the $Ag_2Se$ film is in fact a porous network intertwined with numerous $Ag_2Se$ nanograins sintered from large aspect ratio of $Ag_2Se$ nanowires; porous structure can accommodate the bending of the film[36], hence, the film itself should have a certain flexibility. Moreover, the nylon membrane and the $Ag_2Se$ film have a good bonding (see Fig. 3b, c). In addition, most recently, Shi et al.[37] reported that $Ag_2S$ semiconductor exhibits an extraordinary metal-like ductility with high plastic deformation strains at room temperature. As Se and S are in the same family, $Ag_2Se$ may have a similar ductility.

**Device performance**. As shown in Fig. 4a, the TE prototype device consists of 4 pieces of the film. Each leg is 5 mm in width and 20 mm in length. To decrease the contact resistance between the film and silver paste[38,39], gold was first evaporated on two ends of each leg and then silver paste was painted to connect the legs in series. The open-circuit voltage and output power were measured with a homemade apparatus (see Supplementary Fig. 11 and Supplementary Note 6). Figure 4b shows the relationship between the open-circuit voltage and temperature difference. The open-circuit voltage is proportional to the temperature difference. When the temperature difference is 30 K, the measured open-circuit voltage is about 18 mV. And the calculated open-circuit voltage can be estimated by the expression of $U_{oc} = N \cdot |\alpha| \cdot \Delta T$ ($N$ is the number of legs). In this case, $|\alpha|$ is about 143 μV $K^{-1}$, thus the $U_{oc}$ ($= 4 \times 143 \times 10^{-3} \times 30$) is about 17.2 mV, which is quite close to the measured one (the difference may be resulted from the error of the temperature (±1 K) during the measurement). The output power ($P$) is calculated by the equation as follows:

$$P = I^2 R_{load} = \left( \frac{U_{oc}}{R_{in} + R_{load}} \right)^2 R_{load} \qquad (1)$$

where $I$ is the output current, $R_{load}$ is the load resistance and $R_{in}$ is the internal resistance of the device.

The curve of output voltage—current and that of output power—current are shown in Fig. 4c. The output voltage is inversely proportional to output current. At the temperature difference of 30 K, the measured maximum output power is about 460 nW. According to Eq. (1), when the load resistance ($R_{load}$) equals to the internal resistance ($R_{in}$) of the device, the maximum output power is obtained. And the load resistance which can be calculated from the Fig. 4c is about 250 Ω (the voltage is about 10.7 mV and the current is about 42.9 μA), matches the measurement of the internal resistance of the device. The maximum power density is about 2.3 W $m^{-2}$, obtained from dividing the power by the cross-sectional area and the number of legs[18], which is somewhat higher than that of reported flexible n-type devices[40–42], confirming that the $Ag_2Se$ film possesses high TE performance.

## Discussion

In summary, we used a simple and low cost process compared with other published methods, i.e., first synthesis of $Ag_2Se$ nanowires then vacuum assisted filtration on nylon membrane and finally hot pressing at a relatively low temperature, to endow the film starting from the $Ag_2Se$ nanowires both good TE

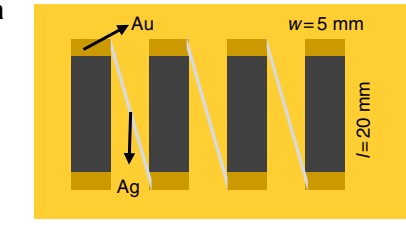

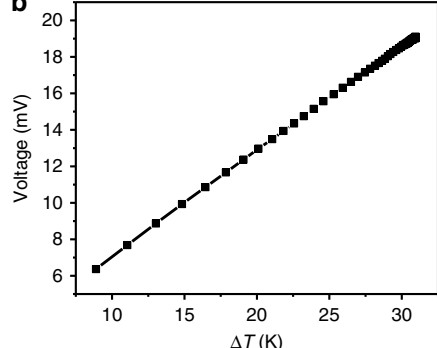

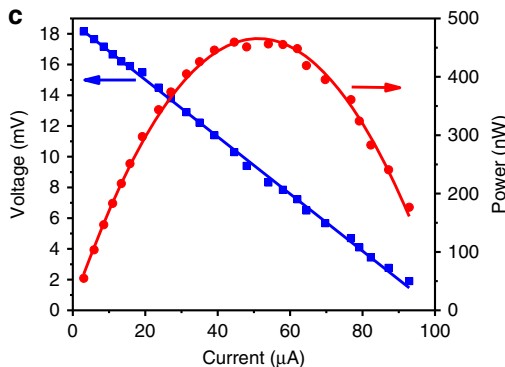

**Fig. 4** Schematic illustration and performance of the prepared device. **a** The schematic illustration of the TE device. **b** The open-circuit voltage at various temperature difference. **c** The output voltage and output power versus current at temperature difference of 30 K

properties and excellent flexibility. A high power factor of 987.4 ± 104.1 μW $m^{-1}$ $K^{-2}$ at 300 K was obtained, which is a record value among the recently reported n-type flexible TE materials. The high power factor comes from the special $Ag_2Se$ film; and the excellent flexibility comes from the nylon membrane, the porous nanostructured $Ag_2Se$ film, and good combination of the $Ag_2Se$ film and nylon membrane. A prototype device with 4-leg of the film connected with silver paste was fabricated. The maximum power density of the device is about 2.3 W $m^{-2}$ at a temperature difference of 30 K. Together with the good TE properties of the $Ag_2Se$ film and excellent flexibility of the nylon membrane, the $Ag_2Se$ film on nylon membrane has shown great promise in flexible TE modules for wearable energy harvesting. This work demonstrates an effective route to fabricate high-performance flexible TE films.

## Methods

**Synthesis**. The $Ag_2Se$ film was prepared by a vacuum-assisted filtration of a $Ag_2Se$-nanowire dispersion on a porous nylon membrane and the film on the nylon membrane was dried at 60 °C in vacuum overnight and then hot-pressed at 200 °C and 1 MPa for 30 min. See details in Supplementary Figs. 1–4 and Supplementary Note 1–2. The bulk $Ag_2Se$ was hot-pressed at 200 °C and 24 MPa for 30 min for comparison (Supplementary Note 3).

**Characterization**. The phase composition of the $Ag_2Se$ film was examined by X-ray diffraction (XRD) using Cu Kα radiation (D/MAX 2550VB3+/PCII). The

morphology of the film was observed by a field emission scanning electron microscope (FESEM, FEI Nova NanoSEM 450). The internal microstructure of the film was examined by double-aberration corrected transmission electron microscope (TEM, FEI Titan @300 kV in TEM and STEM mode), and the TEM sample was prepared by the Focused Ion Beam (FIB, FEI Helios600i) with the in-situ lift-out technique. To protect the sample surface before the ion milling, a Pt layer was sputtered on the full sample. The region of interest was further locally capped in the FIB with ion beam deposited carbon. The major milling was done with a 30 kV Ga ion beam while the milling progress was controlled with the scanning electron microscope. Final milling to minimize the damage layer on the specimen was performed with 5 kV followed by 2 kV Ga ion beam. X-ray photoelectron spectroscopy (XPS, ESCALAB 250Xi) was used to examine the bonding energy of the film. The pore distribution and pore volume were measured by a Brunauer-Emmett-Teller (BET) analyzer (Micromeritics, ASAP 2020).

**Measurement of TE properties and performance**. The in-plane electrical conductivity and Seebeck coefficient were measured by the standard four-probe method (Sinkuriko, ZEM-3) in He atmosphere. The in-plane thermal conductivity was calculated by $\kappa = \rho \cdot D \cdot C_p$, where $\rho$, $D$, and $C_p$ are the density, thermal diffusivity, and specific heat capacity, respectively. The in-plane $D$ was measured by the laser flash method (LFA467, NETZSCH), $C_p$ was measured by the differential scanning calorimetry (DSC Q2000, TA), and the $\rho$ was obtained by measuring the mass and geometrical dimensions of the film with nylon (Supplementary Note 4). The Hall coefficient was measured by the Hall measurement system (LakeShore 8404). The measurement error for $\sigma$ and $\alpha$ is about ±5%. The thickness of the films was determined by a thickness meter (Shanghai Liu Ling Instrument Factory) combined with FESEM observation. The bending test of the film was performed using a homemade apparatus around a rod with a diameter of 8 mm.

The film was cut into strips (20 mm × 5 mm), and the strips were pasted on a polyimide substrate (the interval of two strips is ~5 mm). The two ends of each strip were coat with a layer of Au via a mask and evaporation. After that, each strip was connected in series with Ag paste as conductive connection to obtain a prototype power generator. The output voltage and output power of the device were measured by a homemade apparatus (see Supplementary Fig. 11 and Supplementary Note 6).

## Data availability
The data that support the findings of this study are available from the corresponding author upon reasonable request.

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

## Acknowledgements

This work was supported by the Key Program of National Natural Science Foundation of china (51632010 and 51632005), National Basic Research Program of China (973 Program) under Grant No. 2013CB632500, the foundation of the State Key Lab of Advanced Technology for Material Synthesis and Processing (Wuhan University of Technology), and Innovation Commission of Shenzhen Municipality (Grant Nos. KQTD2016022619565991 and KQCX2015033110182370).

## Author contributions

Y.D. and K.C. conceived the idea, discussed and analyzed the data. Y.D. performed the majority of experiments and drafted the manuscript; K.C. designed the whole work and revised the manuscript; Y.Q and J.H. prepared the TEM sample and analyzed the microstructure by STEM; Q.Y. and L.C. contributed the thermoelectric properties measurement; L.C. proposed valuable advice for revising the manuscript. S.C. contributed the theoretical calculation. All authors discussed the results and commented on the manuscript.

## Additional information

**Competing interests:** The authors declare no competing interests.

