## [Peer Review File · Nature Communications]

Reviewers' comments:

Reviewer #1 (Remarks to the Author):

This paper demonstrates high-performance flexible TE films made by vacuum filtration and hot press. While the results are interesting, a major revision is needed to address the following issues before this work can be published on Nature Communications.

The films were fabricated by vacuum filtration and hot press. However, the power factor of the film is still much lower than the bulk Ag₂Se fabricated by hot press in literature. The author should provide explanations. I suggest the author make a bulk pellet using their Ag₂Se nanowires as a reference sample and compare the TE properties between the film and the bulk pellet.

The figure 2 does not show electrical conductivity above 400 K. The authors should provide better explanations of the electrical and Seebeck trend vs. temperature. I suggest the author perform Hall measurement to provide carrier concentration and mobility changes with temperature.

While it is indeed not easy to measure in-plane thermal conductivity, such measurement can be done on thick films with carefully designed measurement setup. The authors should attempt to provide thermal conductivity data and compared with bulk values.

In figure 4b, the power measurement should be done with more uniform current distributions. There is no measurement for current in the 45-65 μ A range, in which the maximum power occurs. It is also useful to compare the voltage and power measurement vs. some calculation based on the TE properties of the films. The power measurement can provide some validation for the measured TE properties.

The nylon membrane only provides a substrate for the TE film. It can be misleading saying it is "inorganic/organic hybrid TE films".

Reviewer #2 (Remarks to the Author):

"High performance n-type Ag₂Se film on nylon membrane for flexible thermoelectric power generator" is a paper that reports on the thermoelectric performance and mechanical properties of Ag₂Se films manufactured by hot pressing Ag₂Se nanowire dispersions onto a porous nylon membrane.

The films show very good room temperature thermoelectric performance and the good mechanical stability when subjected to bending cycles. The authors have demonstrated one of the highest power factor, flexible n-type materials seen so far in the research literature. The paper is concise and written clearly.

The paper could be improved if the following discussion points were taken into account:

1. The authors should include a short discussion on the chemical stability of the films i.e. are they subject to oxidation during processing and how might this effect thermoelectric performance. If the authors have carried out any EDX or XPS analysis that may give an indication of the formation of Na₂SeO₃ and Na₂SeO₄, these should be given.
2. The authors should expand their discussion on thermal conductivity as related to porosity and provide a quantification of pore volume and distribution by, for example Brunauer–Emmett–Teller (BET) and Barrett–Joyner–Halenda (BJH) analysis. This may lead to a better understanding of how the porosity effects the lattice contribution to thermal conductivity.

Reviewer #3 (Remarks to the Author):

The paper reports the thermoelectric properties of Ag₂Se film/nylon membrane that are mechanically flexible. High power factor was obtained and the prototype four unileg module showed rather high output power density. Although the results look good, the paper doesn't meet the criteria of Nat. Commun. in terms of the originality of the idea and the impact to the thermoelectric community as well as other scientific communities. It is suggested that the authors should submit the paper to more specified journals. Some of the reasons for the above comments are as follows:

- *Synthetic process of Ag₂Se nanowires is already known and the authors only followed it.

Characterization and properties measurements are standard ones. *The obtained power factor at room temperature is ~980 mW/mK² which is among the highest value for n-type flexible TE materials, but it is far lower than the values reported for bulks or films. It is quite reasonable because the present film contains nanoparticles with many grain boundaries even though nanowires were employed as starting materials and a certain amount of pores which would interrupt smooth electronic transport. It would imply that the nano/micro structure has not been optimized and there is still room for further improvement in TE properties, so that high power factor (the authors insist) obtained here is not so extraordinary. *Flexibility of the present film materials solely depends on that of nylon. This has been also the case for other flexible TE materials based on inorganic/organic composites such as Bi₂Te₃/polymer, so that it is not surprising if the present material/device is flexible.

We thank the reviewers very much for their valuable and helpful comments. We have tried our best to answer the reviewers' questions in detail. Please find below our detailed response to each comment of the referee in red, along with a description of any subsequent text modification in the revised manuscript in purple.

Reviewer #1 (Remarks to the Author):

This paper demonstrates high-performance flexible TE films made by vacuum filtration and hot press. While the results are interesting, a major revision is needed to address the following issues before this work can be published on Nature Communications.

Response: We thank the reviewer #1 very much for his/her positive comments. Hereby, we try our best to resolve the problems one by one.

The films were fabricated by vacuum filtration and hot press. However, the power factor of the film is still much lower than the bulk AgSe fabricated by hot press in literature. The author should provide explanations. I suggest the author make a bulk pellet using their AgSe nanowires as a reference sample and compare the TE properties between the film and the bulk pellet.

RESPONSE: Thanks for this suggestion. We have hot pressed a bulk Ag₂Se at 200 °C and 24 MPa for 30 min. The TE properties of the bulk sample are shown in Fig.R1. It can be seen that the change tendency with temperature of all the parameters is quite similar to that of our film. At room temperature the Seebeck coefficient of the bulk is similar to that of the film, whereas the electrical conductivity (~629 S/cm) is larger than that of the film. As a result, the power factor of the bulk Ag₂Se (about 1340 μWm⁻¹ K⁻²) is also larger than that of the film.

Fig.R1. TE properties of the bulk Ag_2Se . Temperature dependence of the Seebeck coefficient, electrical conductivity and power factor for the bulk Ag_2Se .

The cross-section FESEM image of the bulk sample shows a large number of nano- and micro-sized pores (Fig. R2 left), which is in good agreement with the low relative mass density of $\sim 66\%$ (density= 5.44 g/cm^3). The relative mass density of the bulk is very similar to the film sample (70%, determined by the mass and geometrical dimensions of the film, and the mass of the film was determined by measuring the film on nylon membrane and another nylon with the same size).

Fig. R2 Cross-section FESEM image (left) and XRD pattern (right) of the bulk Ag_2Se

The bulk sample was also examined with XRD (shown in Fig. R2 right). It does not show preferential orientation along (00 l), which is different from the film sample (We deduce that the vacuum assisted filtration process for the film probably has an effect to let the Ag_2Se nanowires orderly stack on the nylon membrane). Theoretical calculations indicate that the (00 l) direction is not a good direction for TE performance, therefore the bulk sample with random orientation grains has higher TE properties.

For the bulk samples prepared by melting high purity elements at 1273 K for 10 h reported in ref. [1], three samples from one ingot, AS1, AS2, and AS3 with different residual carrier concentrations were selected with power factor of 1850, 3520, and 1060 $\mu\text{Wm}^{-1}\text{K}^{-2}$, respectively. The AS2 has the highest power factor due to having substantially higher carrier mobility and slightly lower carrier concentration. Hence, we think that the main reasons for the lower power factor of our film are: (1) high porosity, (2) the grains preferentially grown along (00l), and (3) the carrier concentration not optimized.

We made some changes in the second paragraph of page 5 in our revised manuscript:

“However, it is still much lower than the best value of the bulk Ag_2Se reported in ref. [27]..... and its carrier concentration not optimized.”

The figure 2 does not show electrical conductivity above 400 K. The authors should provide better explanations of the electrical and Seebeck trend vs. temperature. I suggest the author perform Hall measurement to provide carrier concentration and mobility changes with temperature.

RESPONSE: Actually, we did have the data above 400 K. The figure with full data is shown as in Figure R3.

Figure R3 In-plane TE properties of the film. Temperature dependence of Seebeck coefficient, electrical conductivity and power factor for the Ag_2Se film.

The Hall measurement results are shown in Fig. R4: the carrier concentration increases from $3.19\text{E}+18$ to $8.46\text{E}+18\text{ cm}^{-3}$ with temperature increasing and keeps constant above 420 K. The mobility decreases with temperature increasing and shows a sharp decrease around the transition temperature, 407 K. The temperature dependence of the Seebeck coefficient and electrical

conductivity of the film can be understood from the Hall measurement results, since the Seebeck coefficient is inversely proportional to the power of 2/3 of the carrier concentration² and the electrical conductivity, carrier concentration (n) and carrier mobility (μ) have the relation: $\sigma = ne\mu$.

Fig. R4 Temperature dependence of carrier concentration and mobility for the Ag₂Se film.

We added the Hall measurement result as Figure 2b and made some changes in the second paragraph of Page 4 in our revised manuscript:

“Figure 2(a) and (b) exhibit the TE properties and Hall measurement results of the Ag₂Se film from 300 to 453 K, respectively. the carrier concentration and the electrical conductivity, carrier concentration (n) and carrier mobility (μ) have the relation: $\sigma=ne\mu$.”

We made some changes in the first paragraph of Page 5 in our revised manuscript:

“However, the change tendency above 393 K of the electrical conductivity of the film is not as that for the Ag₂Se film. while that of the latter only drops from ~600 to 500 cm²/Vs .”

While it is indeed not easy to measure in-plane thermal conductivity, such measurement can be done on thick films with carefully designed measurement setup. The authors should attempt to provide thermal conductivity data and compared with bulk values.

RESPONSE: Good point and very challenge. In order to gain the in-plane thermal conductivity of the film, we directly measure the in-pane thermal conductivity of the Ag₂Se film with nylon membrane as suggested by NETZSCH Group due to difficulty of removing nylon membrane. Hence, Laser-Flash (NETZSCH LFA-467) method was used to measure the in-plane thermal

diffusivity (D). To determine the thermal conductivity (κ), the density (ρ) and the specific heat capacity at constant pressure (C_p) were also measured to calculate the $\kappa = \rho C_p D$. C_p was measured by the differential scanning calorimetry (DSC Q2000, TA), and the ρ was obtained by measuring the mass and geometrical dimensions of the film with nylon. The results at room temperature are shown in Table. R1.

Table.R1 The parameters of in-plane thermal conductivity of the Ag_2Se film with nylon at room temperature.

Thermal diffusivity s^{-1}	Specific heat capacity $\text{mm}^2 \text{J g}^{-1} \text{K}^{-1}$	Density g cm^{-3}	Thermal conductivity $\text{W m}^{-1} \text{K}^{-1}$
0.50 ± 0.006	0.75 ± 0.02	1.197 ± 0.05	0.449 ± 0.03

Compared with the thermal conductivity of bulk Ag_2Se ($0.6\text{-}1.2 \text{ Wm}^{-1}\text{K}^{-1}$)³⁻⁶, the thermal conductivity of the film with nylon is somewhat lower. This is very reasonable. As the hybrid film contains nylon membrane (with $\kappa \sim 0.25\text{-}0.36 \text{ Wm}^{-1}\text{K}^{-1}$)⁷ and Ag_2Se film, which contains pores with sizes ranging from $\sim 20 \text{ nm}$ to submicrometers (As shown in FESEM image of Figure 1c in the manuscript and BET result (Fig. R7 hereinafter)), nanograins and dislocations, and moreover, there is a hetero-interface between the Ag_2Se film and nylon membrane.

As we only measured the in-plane thermal conductivity of the film with nylon, and we measured the pore size distribution (required by the reviewer #2), we made some changes in Page 9 in our revised manuscript:

“As it is hardly to separate the Ag_2Se film from the nylon membrane without destroy the film, ...

.... $0.478 \text{ W m}^{-1} \text{K}^{-1}$. Hence, the ZT value at 300 K of our film will be in the range of 0.62 to 0.66.”

In figure 4b, the power measurement should be done with more uniform current distributions. There is no measurement for current in the 45-65 μA range, in which the maximum power occurs. It is also useful to compare the voltage and power measurement vs. some calculation based on the TE properties of the films. The power measurement can provide some validation for the measured TE

properties.

RESPONSE: Thanks for this suggestion. The data for current in the 45-65 μA range have been added and the figure has been re-plotted (see Fig. R5b).

The comparison of the measured voltage and calculated one based on the TE properties is as follows: for example, when the temperature difference is 30 K, the measured open-circuit voltage is about 18 mV. And the calculated open-circuit voltage can be estimated by the expression of $U_{oc}=N\cdot|\alpha|\cdot\Delta T$ (N is the number of legs). In this case, $|\alpha|$ is about $143 \mu\text{V K}^{-1}$, thus the result of U_{oc} ($=4 \times 143 \times 10^{-3} \times 30$) is about 17.2 mV, which is quite close to the measured one (the difference may be resulted from the error of the temperature (± 1 K) during the measurement).

The curve of output voltage - current and that of output power – current are shown in Fig. R5(b). The output voltage is inversely proportional to output current. When the load resistance equals to the internal resistance (R_{in}) of the device, the maximum output power is obtained. The output power (P) is calculated by the equation as follows:

$$P = \frac{U^2}{4R_{in}} = IU \quad (1).$$

At the temperature difference of 30 K, the measured maximum output power is about 460 nW and the applied load resistance is 240Ω . The internal resistance which can be calculated from the figure R5(b) is about 250Ω (the voltage is about 10.7 mV and the current is about $42.9 \mu\text{A}$), which matches the sum of the load resistance of the device and the internal resistance of the ammeter ($\sim 10 \Omega$).

Fig. R5 Performance of the designed device, (a) the open-circuit voltage at various temperature difference (inset is a digital photo of the TE prototype device), (b) the output voltage and output power versus current at a temperature difference of 30 K.

We changed Figure 4b in page 8. And we added some sentences in the third paragraph of page 7 in the revised version:

“When the temperature difference is 30 K, which is quite close to the measured one (the difference may be resulted from the error of the temperature (± 1 K) during the measurement).”

And we added some sentences in the first paragraph of page 8 in the revised version:

“At the temperature difference of 30 K, the measured maximum output power is about 460 nW and the applied load resistance is 240 Ω, which matches the sum of the load resistance of the device and the inner resistance of the ammeter (~ 10 Ω).”

The nylon membrane only provides a substrate for the TE film. It can be misleading saying it is “inorganic/organic hybrid TE films”.

RESPONSE: We have modified the description in the revised version.

Reviewer #2 (Remarks to the Author):

“High performance n-type Ag₂Se film on nylon membrane for flexible thermoelectric power generator” is a paper that reports on the thermoelectric performance and mechanical properties of Ag₂Se films manufactured by hot pressing Ag₂Se nanowire dispersions onto a porous nylon membrane.

The films show very good room temperature thermoelectric performance and the good mechanical stability when subjected to bending cycles. The authors have demonstrated one of the highest power factor, flexible n-type materials seen so far in the research literature. The paper is concise and written clearly.

We thank the reviewer #2 very much for his/her positive comments. Hereby, we try our best to resolve the problems one by one.

The paper could be improved if the following discussion points were taken into account:

1. The authors should include a short discussion on the chemical stability of the films i.e. are they

subject to oxidation during processing and how might this effect thermoelectric performance. If the authors have carried out any EDX or XPS analysis that may give an indication of the formation of Na_2SeO_3 and Na_2SeO_4 , these should be given.

Response: Thanks for this suggestion. We have carried out XPS analysis of the film. As shown in the Fig. R6, the binding energies of Se 3d and Ag 3d_{5/2} are 54.16 eV and 367.86 eV (the standard data of Ag 3d_{5/2} in Ag₂Se is 367.8 eV), respectively, indicating that these two elements exist in the film as Ag⁺ and Se²⁻. And there are no other peaks about Se and Ag (The standard data: Ag₂O 356.6 eV), it suggests that no oxidation like Na₂SeO₃ and Na₂SeO₄ formed during the processing. The O1s and C1s peaks in the survey spectrum should be due to from the nylon membrane.

Fig. R6 The XPS spectra of the Ag₂Se film: (a) XPS survey spectrum; (b) Se 3d spectra; (c) Ag 3d spectra.

Hence, in the revised version, we added the following words in the first paragraph of page 4:

“XPS analyses (see supplementary Fig. 9) indicate that the film is stable and no oxidation occurred during the processing.”

2. The authors should expand their discussion on thermal conductivity as related to porosity and provide a quantification of pore volume and distribution by, for example Brunauer–Emmett–Teller (BET) and Barrett–Joyner–Halenda (BJH) analysis. This may lead to a better understanding of how the porosity effects the lattice contribution to thermal conductivity.

Response: We have measured the film on nylon by the BET and BJH methods. The pore

distribution and pore volume of the Ag₂Se film are shown in Fig.R7.

Fig. R7 Pore distribution and pore volume for the Ag₂Se film on nylon.

It can be seen that the pore size is in the range of ~ 20 nm to 240 nm (the pores with size around 220 nm should be mainly from the nylon membrane, the majority of pores in the film is in the range of ~90- 200 nm). Combined with the result of FESEM observation, the pores are in the range of ~20 nm to submicron, and they are irregular and randomly distributed. Therefore, the presence of disorder in terms of nanocrystallite boundaries, microstructural defects (dislocations and amorphous regions) and nanoporous in our nanostructured Ag₂Se sample, covers a broad dimensional nano-to-meso scale range to scatter heat-carrying phonons in a wide spectrum of wavelengths. The microstructural defects, such as dislocations (a few nm) and heterointerfaces are effective in scattering the short wavelength phonons, the nanocrystallite boundaries and residual nanopores (~20–200 nm) play an effective role in scattering the mid-to-high wavelength phonons^{8–10}, thus leading to a very low value of κ .

We added some sentences in the second paragraph of Page 9 in our revised manuscript::

“.....As our Ag₂Se film contains pores with sizes ranging from ~20 nm to submicron.... in scattering the mid-to-high wavelength phonons.....”

Reviewer 3

The paper reports the thermoelectric properties of Ag₂Se film/nylon membrane that are mechanically flexible. High power factor was obtained and the prototype four unileg module showed rather high output power density. Although the results look good, the paper doesn't meet the criteria of Nat. Commun. in terms of the originality of the idea and the impact to the

thermoelectric community as well as other scientific communities.

It is suggested that the authors should submit the paper to more specified journals.

Some of the reasons for the above comments are as follows:

*Synthetic process of Ag₂Se nanowires is already known and the authors only followed it. Characterization and properties measurements are standard ones.

Response: We thank reviewer #3 a lot for his/her careful reviews and suggestive comments, which we do think can help us strengthen our work. As for synthetic process, we would like to mention that we here used a simple and low cost process compared with other published methods, i.e., first synthesis of Ag₂Se nanowires then vacuum assisted filtration on nylon membrane and finally hot pressing at a relatively low temperature, to endow the film starting from the Ag₂Se nanowires both good thermoelectric properties and excellent flexibility. We think this is a very good new idea!

*The obtained power factor at room temperature is ~980 mW/mK² which is among the highest value for n-type flexible TE materials, but it is far lower than the values reported for bulks or films. It is quite reasonable because the present film contains nanoparticles with many grain boundaries even though nanowires were employed as starting materials and a certain amount of pores which would interrupt smooth electronic transport. It would imply that the nano/micro structure has not been optimized and there is still room for further improvement in TE properties, so that high power factor (the authors insist) obtained here is not so extraordinary.

Response: Yes, the power factor at room temperature ~980 μW/mK² is lower than that the highest value for Ag₂Se bulks. As you can see details in our response to reviewer #1, the highest reported value (3520 μW/mK²) of the bulk was prepared by melting high-purity elements at 1273 K for 10 h [1], obviously it is high energy cost. Moreover, in ref. [1], the prepared ingot is not homogeneous, besides the sample with the high power factor, another selected sample shows a power factor of ~1060 μW/mK², which is close to that of our film and much lower than that (1380 μW/mK²) of our bulk sample hot pressed at 200 °C and 24 MPa for 30 min, which we did the additional experiments required by the reviewer #1.

It is a common phenomenon that nanostructures are quite difficult to be highly densified. The

porous structure will decrease the electrical conductivity, however, it also can decrease the thermal conductivity, and on the other hand, the porous structure can accommodate the bending of the film¹¹.

As for the reason for the lower power factor, we agree with the reviewer's point. And we think that there are other reasons, such as the grains preferentially grown along (00*l*) and the carrier concentration not optimized. It will be our future work to further improve the power factor of the film.

*Flexibility of the present film materials solely depends on that of nylon. This has been also the case for other flexible TE materials based on inorganic/organic composites such as Bi₂Te₃/polymer, so that it is not surprising if the present material/device is flexible.

Response: Thank the reviewer for the comment. Yes, nylon plays a very important role for the excellent flexibility. In addition, we think there are other two important factors: one is the intertwined porous nanograin network (porous structure can accommodate bending of the film¹¹), and another one is the good combination between the nanograin network and the nylon membrane. Lack of any of them, there would have no such excellent flexibility.

There have some works reported on flexible TE materials based on inorganic/organic composites such as Bi₂Te₃/polymer. For example: Gao et al.¹² reported a novel glass-fiber-aided cold-pressing method for flexible n-type Ag₂Te films on copy paper and the power factor value was 85 μW m⁻¹ K⁻² at 300 K. The electrical conductivity is 80% of the original value after 500 times bending around a rod with 5 mm radius. Jin et al.¹³ deposited n-type Bi₂Te₃ thick film on a cellulose fibers paper via an unbalanced magnetron sputtering technique and the film exhibited a power factor of ~250 μW m⁻¹ K⁻² at room temperature. After 100 times bending around a rod with radius of 10 mm, the resistance of the film increased by ~6%. And most recently, Wang et al.¹⁴ fabricated p-type PEDOT/Bi₂Te₃ hybrid films with monodispersed and periodic Bi₂Te₃ nanophase via a very complicated process. An optimized power factor of ~1350 μW m⁻¹ K⁻² was obtained. After 100 times bending around a rod with radius of 3.5 mm, the resistance of the film increased by ~4%. Compared with these reported inorganic/organic composite films, our film shows better flexibility. In general, we prepared Ag₂Se film on nylon membrane by a very simple method, and the hybrid

film shows very good thermoelectric properties and excellent flexibility.

Hence, in the revised version, we added the comparison of the flexibility in the last paragraph of page 5 to the first paragraph of page 6:

“About 93% and 80% of the initial electrical conductivity are maintained after 1000 and 1500 bending cycles, respectively., our film shows a better flexibility (see supplementary Table 2).”

We hope that the revised version will be acceptable for publication in *Nature Communications*

Kefeng Cai

References

1. Ferhat, M. & Nagao, J. Thermoelectric and transport properties of β -Ag₂Se compounds. *J. Appl. Phys.* **88**, 813–816 (2000).
2. Snyder, G. J. & Toberer, E. S. Complex thermoelectric materials. *Nat. Mater.* **7**, 105–114 (2008).
3. Mi, W. *et al.* Thermoelectric transport of Se-rich Ag₂Se in normal phases and phase transitions. **133903**, 0–5 (2016).
4. Wang, H. *et al.* Low-temperature thermoelectric properties of b-Ag₂Se synthesized by hydrothermal reaction. *J. Electron. Mater.* **40**, 624–628 (2011).
5. Lee, C., Park, Y. & Hashimoto, H. Effect of nonstoichiometry on the thermoelectric properties of a Ag₂Se alloy prepared by a mechanical alloying process. *J. Appl. Phys.* **101**, 24920 (2007).
6. Xiao, C. Synthesis and optimization of chalcogenides quantum dots thermoelectric materials. *Springer Berlin Heidelberg*. (2016).
7. King, J. A., Tucker, K. W., Vogt, B. D., Weber, E. H. & Quan, C. Electrically and thermally conductive nylon 6,6. *Polym. Compos.* **20**, 643–654 (1999).
8. Bux, B. S. K. *et al.* Nanostructured bulk silicon as an effective thermoelectric material. *Adv. Funct. Mater.* **19**, 2445–2452 (2009).
9. Bathula, S. *et al.* Enhanced thermoelectric figure-of-merit in spark plasma sintered nanostructured n-type SiGe alloys. *Appl. Phys. Lett.* **101**, 213902 (2012).
10. Zhu, G. H. *et al.* Increased phonon scattering by nanograins and point defects in nanostructured silicon with a low concentration of germanium. *Phys. Rev. Lett.* **102**, 196803 (2009).
11. Yang, W. *et al.* Large-deformation and high-strength amorphous porous carbon nanospheres. *Sci. Rep.* **6**, 24187 (2016).
12. Gao, J. *et al.* A novel glass-fiber-aided cold-press method for fabrication of n-type Ag₂Te nanowires. *J. Mater. Chem. A* **5**, 24740–24748 (2017).
13. Jin, Q. *et al.* Cellulose fiber-based hierarchical porous bismuth telluride for high-performance flexible and tailorable thermoelectrics. *ACS Appl. Mater. Interfaces* **10**, 1743–1751 (2018).
14. Wang, L., Zhang, Z., Liu, Y., Wang, B. & Wang, S. Exceptional thermoelectric properties of flexible organic –

inorganic hybrids with monodispersed and periodic nanophase. *Nat. Commun.* **9**, 3817 (2018).

Reviewers' comments:

Reviewer #1 (Remarks to the Author):

The authors need to address the following key issues before it can be published.

The Hall measurement result is questionable. The author should explain the trend. Why the carrier concentration first increases with temperature, and then suddenly remain constant? Also, why the mobility suddenly drops significantly at certain temperature?

Laser flash method is applicable to measure thermal diffusivity of bulk samples. Measuring films' thermal diffusivity using this method may involve large errors. The author should explain how they measure the in-plane thermal diffusivity using laser flash.

The nylon is only the substrate, and is not part of the TE film. The author cannot claim it is a hybrid film. Therefore, it is not valid to use the total thermal conductivity of the TE film and nylon substrate to represent the thermal conductivity of the TE film. The author needs to obtain the thermal conductivity of the TE film in order to calculate the ZT. Keep in mind the electrical and Seebeck are measured on the TE film, and the substrate does not play any role in the electrical and Seebeck measurement result.

Lastly, the author should better highlight the true novelty of their work compared with the work that has been done in literature.

Reviewer #2 (Remarks to the Author):

I believe the comments have been addressed sufficiently and am satisfied with the authors' changes to the manuscript.

Reviewer #3 (Remarks to the Author):

The authors perfectly revised the manuscript according to the reviewers' comments by including new experimental results and detailed discussion. It is judged that it is suitable for publication in Nat. Commun.

We thank the reviewers very much for the comments. Those comments are very helpful for further improving our manuscript. We have tried our best to answer the reviewers' questions, and also made some modifications in the updated manuscript. Please find below our detailed response to each comment of the referee in red, along with a description of any subsequent text modification in the revised manuscript in purple.

Reviewers' comments:

Reviewer #1 (Remarks to the Author):

1. The authors need to address the following key issues before it can be published.

The Hall measurement result is questionable. The author should explain the trend. Why the carrier concentration first increases with temperature, and then suddenly remain constant? Also, why the mobility suddenly drops significantly at certain temperature?

Response: Thank the reviewer #1 very much for positive comment and the additional questions. In fact the temperature dependence of the carrier concentration and the mobility of our film is quite similar to that of the bulk Ag_2Se reported in ref. [1] (see Fig.1R below). The carrier concentration first increases with increasing temperature, which is due to thermal excitation of carriers (the concurrent decrease in the Seebeck coefficient of the film suggests that the additional carriers are holes). When the temperature is beyond the phase transition temperature (407 K), the Ag_2Se becomes a superionic conductor, and then the carrier concentration almost keeps constant. The mobility suddenly drops significantly across the superionic transition, the reason for which is that the Ag ions in superionic Ag_2Se move freely and tend to scatter electrons more efficiently than a static lattice [1].

Fig. 1R Transport properties of Ag_{2+x}Se . Circles represent hot-pressed samples, squares represent samples cut from ingots (ref. 1)

Hence, in the revised manuscript (first paragraph of page 5) we revised the related description as:

“The temperature dependence of the carrier concentration and the mobility of our film is quite similar to.... the Ag ions move freely and tend to scatter electrons more efficiently than a static lattice [24].”

2. Laser flash method is applicable to measure thermal diffusivity of bulk samples.

Measuring films' thermal diffusivity using this method may involve large errors. The author should explain how they measure the in-plane thermal diffusivity using laser flash.

The nylon is only the substrate, and is not part of the TE film. The author cannot claim it is a hybrid film. Therefore, it is not valid to use the total thermal conductivity of the TE film and nylon substrate to represent the thermal conductivity of the TE film. The author needs to obtain the thermal conductivity of the TE film in order to calculate the ZT. Keep in mind the electrical and Seebeck are measured on the TE film, and the substrate does not play any role in the electrical and Seebeck measurement result.

RESPONSE: The detailed laser flash measurement of in-plane thermal diffusivity of our films is explained as follows:

First step is making a film disc with diameter of ~ 25 mm; second step is digging a hole with the size of ~ 5 mm in the center of the film; last step is measuring the

in-plane diffusivity of the film by heating the center part around the hole with laser in NETZSCH LFA-467. Repeat the measurement with a larger hole-size of 10 and 12 mm and then obtain the in-plane diffusivity by averaging the three test results. The system error of the measurement mainly depends on the thermal property and thickness of the measured film. The lower the thermal conductivity of the film is, the larger the measurement error is; and the thicker the film is, the smaller the measurement error is.

For sure, it will be great if the in-plane thermal conductivity of the pure Ag₂Se film can be measured. However, as mentioned above, the challenge is always there not only for us but also for all the researchers in this community. For us, it is hard to separate the Ag₂Se film from the nylon membrane without destroy the film because the size of the film for the measurement is relatively large. Finally, we have to measure the Ag₂Se film with nylon membrane. And we agree with the reviewer #1 that it is not accurate to use the in-plane thermal conductivity to calculate the ZT value. Therefore, considering the measured value has reference value for practical application of the film, we modified the related description in the revised version (first paragraph of page 9) as follows:

“As it is hardly to separate the Ag₂Se film from the nylon membrane without destroy the film, the in-plane thermal conductivity of the Ag₂Se film cannot be provided here (the in-plane thermal conductivity of the Ag₂Se film on nylon membrane was measured to be 0.449 Wm⁻¹K⁻¹, see supplementary Table1). However, it is deduced.....the ZT value at 300 K of our film is estimated to be ~0.6.”

We have modified the description of “hybrid film” in the revised version into “Ag₂Se film on nylon membrane”.

3. Lastly, the author should better highlight the true novelty of their work compared with the work that has been done in literature.

RESPONSE: Thanks for pointing out. The novelty of our work is as follows: we here used a simple and low cost process compared with other published methods, i.e., first

synthesis of Ag₂Se nanowires then vacuum assisted filtration on nylon membrane and finally hot pressing at a relatively low temperature, to endow the film starting from the Ag₂Se nanowires both good thermoelectric properties and excellent flexibility. As listed in Table 1 and the supplementary information (Table S2), the film shows one of the highest power factors among the n-type flexible TE materials and excellent flexibility.

We highlight the novelty of our work in the revised version (in summary of page 10) as follows:

“...we used a simple and low cost process compared with other published methods, i.e., first synthesis of Ag₂Se nanowires then vacuum assisted filtration on nylon membrane and finally hot pressing at a relatively low temperature, to endow the film starting from the Ag₂Se nanowires both good thermoelectric properties and excellent flexibility....”

Reviewer #2 (Remarks to the Author):

I believe the comments have been addressed sufficiently and am satisfied with the authors' changes to the manuscript.

RESPONSE: Thank Reviewer #2 very much for agreeing publication of our manuscript in Nature communications.

Reviewer #3 (Remarks to the Author):

The authors perfectly revised the manuscript according to the reviewers' comments by including new experimental results and detailed discussion. It is judged that it is suitable for publication in Nat. Commun.

RESPONSE: Thank Reviewer #3 very much for agreeing publication of our manuscript in Nature communications.

Reference

1. Day, T. et al. Evaluating the potential for high thermoelectric efficiency of silver selenide. J. Mater. Chem. C 1, 7568–7573 (2013).

REVIEWERS' COMMENTS:

Reviewer #1 (Remarks to the Author):

The authors have addressed the comments. I would recommend it for publication.

REVIEWERS' COMMENTS:

Reviewer #1 (Remarks to the Author):

The authors have addressed the comments. I would recommend it for publication.

RESPONSE: Thank the Reviewer #1 very much for agreeing publication of our manuscript in Nature communications.

Kefeng Cai
